# Hypoxic Regulation of Gene Transcription and Chromatin: Cause and Effect

**DOI:** 10.3390/ijms21218320

**Published:** 2020-11-06

**Authors:** Jessica D. Kindrick, David R. Mole

**Affiliations:** NDM Research Building, Nuffield Department of Medicine, University of Oxford, Roosevelt Drive, Headington, Oxford OX3 7FZ, UK; jessica.kindrick@jesus.ox.ac.uk

**Keywords:** hypoxia, transcription, chromatin, epigenetics, hypoxia-inducible factor, 2-oxoglutarate-dependent dioxygenase, histone, methylation, acetylation

## Abstract

Cellular responses to low oxygen (hypoxia) are fundamental to normal physiology and to the pathology of many common diseases. Hypoxia-inducible factor (HIF) is central to this by enhancing the transcriptional activity of many hundreds of genes. The cellular response to HIF is cell-type-specific and is largely governed by the pre-existing epigenetic landscape. Prior to activation, HIF-binding sites and the promoters of HIF-target genes are already accessible, in contact with each other through chromatin looping and display markers of activity. However, hypoxia also modulates the epigenetic environment, both in parallel to and as a consequence of HIF activation. This occurs through a combination of oxygen-sensitive changes in enzyme activity, transcriptional activation of epigenetic modifiers, and localized recruitment to chromatin by HIF and activated RNApol2. These hypoxic changes in the chromatin environment may both contribute to and occur as a consequence of transcriptional regulation. Nevertheless, they have the capacity to both modulate and extend the transcriptional response to hypoxia.

## 1. Introduction

Cellular responses to reduced levels of oxygen (hypoxia) are fundamental to normal mammalian physiology as well as the pathophysiology of major human diseases. Classically, these include adaptations to the lowered oxygen availability and systemic hypoxia experienced at altitude. However, in evolutionary terms, increases in localized oxygen consumption in actively respiring muscle during exercise have probably driven these responses to a greater extent. Another important cause of physiological hypoxia occurs during growth when it is thought that hypoxia pathways help match the development of new blood vessels to increases in cell number. In addition, many of the diseases prevalent in the Western world also lead to reduced oxygen levels. Congenital heart disease or lung disease, often associated with tobacco smoking, may lead to systemic hypoxia. Anemia will limit oxygen-delivery to the tissues systemically, while atheroma may restrict arterial blood supply leading to localized reduction in oxygen delivery, most notably in heart, brain or limb peripheries and during exercise (e.g., angina). Similarly, increased metabolism in regions of inflammation and infection frequently produce localized areas of hypoxia, which may in turn influence the immune response. Furthermore, the rapid cellular growth seen in many solid cancers may outstrip development of new blood vessels leading to a hypoxic tumor environment and activation of these pathways. In each of these settings, the hypoxic responses generated may be appropriate, aiding adaptation to the pathological insult, or inappropriate, thereby contributing to the pathophysiology of the disease.

To date much of the work on the cellular response to hypoxia has focused on identifying and defining transcriptional responses by examining changes in gene expression and binding of hypoxia-inducible transcription factors to chromatin. However, it is becoming clear that interactions between cis-acting elements of chromatin and trans-acting factors, which include specific transcription factors and non-coding RNA moieties as well as generic components of the transcriptional machinery, are complex. In particular, they involve not just the recognition of specific DNA sequences, but are also influenced by DNA modifications, the addition of particular moieties to specific residues on histones, chromatin accessibility and higher-order chromatin structure. Indeed, it is becoming apparent that hypoxia may also have direct effects on chromatin, in addition to that caused by stabilizing hypoxia-inducible transcription factors. Such effects may be widespread across the genome as a result of altered enzyme abundance or oxygen-dependent activity. Alternatively, they may be targeted to specific loci by hypoxia-inducible transcription factors. This raises the possibility that hypoxia-induced changes in chromatin may further contribute to the transcriptional response to this stimulus. However, the phosphorylation and activation of RNA polymerase 2 during transcription also leads to the recruitment of chromatin-modifying enzymes to transcribing gene loci and so the mere act of transcription itself alters the epigenetic environment. In consequence, the causal relationship for any observed associations between chromatin modification and transcription in hypoxia may operate in either direction. This review will examine the interplay between oxygen availability, the transcriptional response, and the epigenetic landscape.

## 2. Organization of Chromatin Structure

Within the nucleus, DNA is packaged around protein complexes called histones in order to coil into its compact chromosome shape and to assist in unpacking when required. These histones are made up of an octamer of subunits, each having an unstructured amino terminal tail that protrudes from the nucleosome and is subjected to extensive covalent modification such as methylation and acetylation. These “marks” may alter the charge of the histone and influence DNA accessibility or may act as recognition sites for other proteins. In consequence, they have many diverse functional effects on gene expression depending on the type of post-translational modification (PTM) and the specific residue modified. 

Histone modifications are not spread evenly throughout the genome. Instead, PTMs often localize to, and help define, specific regions of a gene. For example, the tri-methylated version of histone 3 lysine 4 (H3K4me3) is most abundant flanking transcriptional start sites (TSS) and “promoters” [1,2]. In contrast, the mono-methylated version, H3K4me1, is found predominantly in “enhancer” regions distant from the TSS [1]. These specific modifications not only localize to particular regions within the gene, but also localize to specific genes based on their expression levels. H3K4me3 is commonly found at active genes, in areas of unwound and relaxed DNA. This open structure allows space for transcriptional machinery to enter, thus generating high levels of gene expression. In contrast, H3K9me3 and H3K27me3 are broadly considered to be repressive marks and are depleted at actively expressed genes. These marks co-localize with tightly condensed areas of chromatin where transcriptional machinery is blocked due to reduced DNA accessibility. While some of these PTMs are relatively stable and defined by the cell type, many others change dynamically throughout the cell cycle and in response to environmental stimuli. 

Enzymes commonly referred to as “writers” or “erasers” are responsible for adding and removing histone marks, respectively. For example, methyl groups are added to histone tails by histone methyltransferases (HMTs) and removed by histone demethylases, whilst acetyl groups are added by histone acetyltransferases (HATs) and removed by histone deacetylases (HDACs). Within each group, individual enzymes are highly specific for particular modification states and residue location. The resulting marks on each histone are a balance between the competing activities acting on each residue and are finely tuned according to genomic location. This is achieved, at least in part, by sequence-specific DNA-binding proteins (e.g., transcription factors), as well as components of the transcriptional complex itself, that help recruit histone-modifying enzymes to particular sites. However, other proteins, referred to as “readers” can recognize specific histone modifications. These may, in turn, recruit additional histone-modifying enzymes, which either “re-write” the modification or add additional modifications in a combinatorial fashion [3].

In addition to histones, DNA itself is also subjected to modification, the best-studied of which is methylation of the cytosine ring in a cytosine-phosphate-guanine (CpG) dinucleotide by the DNA methyltransferases, DNMT1, DNMT3A, and DNMT3B. Around 75% of CpGs in the human genome are methylated, with the majority of unmethylated CpGs being found in CpG islands associated with either gene promoters or within gene bodies [4,5]. Increased methylation at gene promoters is associated with reduced gene expression, but, conversely, methylation within the gene body is often positively correlated with expression [6]. Similar to histone modifications, DNA methylation may be “read” by proteins containing a methyl-CpG-binding domain, which frequently act as transcriptional repressors. Such proteins repress genes by recruiting histone deacetylase activity, which in turn may impact on DNA accessibility [7]. In addition, DNA methylation may interfere with binding of transcription factors to CpG-containing motifs, further inhibiting gene expression [7]. More recently, “eraser” mechanisms to remove CpG-methylation have been identified, catalyzed by the ten-eleven-translocation (TET) enzymes. This involves sequential oxidation of methyl-cytosine to hydroxymethylcytosine, formylcytosine, and carboxylcytosine, potentially leading to regeneration of un-modified cytosine [8].

Finally, the identification of active chromatin regions (from histone/DNA modification, DNA accessibility, transcription factor binding, and RNA analyses) has emphasized the importance of higher-order chromatin structure in gene regulation [9]. Many enhancer regions are 10 s–100 s of kilobases away from the gene promoter on which they operate, with interaction generally thought to occur between multiple enhancers and a given promoter through cohesin-mediated chromatin looping [2,10], although other mechanisms have also been proposed [11,12]. In turn, cohesin may be re-distributed by CCCTC-binding factor (CTCF) in response to transcription [13,14].

## 3. Transcriptional Regulation in Hypoxia

Hypoxia constitutes a major physiological and pathological stimulus that regulates the transcription of many hundreds of genes. These have key roles in pathways that improve oxygen-delivery such as iron-uptake and transport, erythropoiesis, angiogenesis and vascular tone, and that reduce oxygen consumption such as glycolysis, TCA-cycle metabolism, cell proliferation and apoptosis, as well as controlling cell-specific functions [15,16]. The vast majority of these transcriptional changes are regulated by the hypoxia-inducible factor (HIF) family of transcription factors (HIF-1, HIF-2, and HIF-3). HIF-1, 2, and 3 are heterodimeric transcription factors, consisting of an alpha (HIF-1α, HIF-2α, or HIF-3α, respectively) and beta (HIF-1β = ARNT) subunit. HIF-1 and HIF-2 have overlapping but non-redundant roles, resulting, at least in part, from differences in tissue distributions, regulation by hypoxia, chromatin binding, and target gene activation [17,18]. The role of HIF-3 is currently poorly understood and will not be discussed further in this review. The HIF-1β-subunit is constitutively expressed. However, when oxygen is abundant, hydroxylation of two prolyl residues on the α-subunit by the prolyl hydroxylase enzymes (PHD1-3/EGLN1-3) facilitates interaction with the von Hippel-Lindau (VHL) E3 ubiquitin ligase, leading to rapid proteasomal degradation [19,20,21] (Figure 1). These enzymes use molecular oxygen and 2-oxoglutarate as co-substrates, as well as iron and ascorbate as co-factors, and form part of a larger family of 2-oxoglutarate-dependent dioxygenases, which are discussed further below [22,23]. Molecular oxygen is a rate-limiting substrate for this reaction and therefore, in hypoxia, HIF-α is able to escape degradation and heterodimerize with HIF-1β. A third site of hydroxylation on a C-terminal asparagine residue regulates interaction with the p300/CBP transcriptional co-activator and helps regulate transcriptional activity [24,25]. Once the HIF complex is formed, it translocates into the nucleus and binds DNA at a specific motif (RCGTG), referred to as the hypoxia response element (HRE) [26]. 

Although the HRE motif is highly abundant throughout the genome (over 1 million occurrences), ChIP-seq analysis of HIF-binding has identified far fewer binding sites (estimated at 500–1000) [27]. Furthermore, HIF-binding sites vary greatly between different cell-types, indicating that in addition to DNA-sequence, epigenetic factors also play a role in shaping the HIF-response [28,29]. One key factor in defining this binding is DNA-accessibility. High-throughput sequencing analysis of DNase-hypersensitivity (DNase-seq) in normoxic and hypoxic cells reveals that HIF binds exclusively to HRE motifs that lie within regions of DNase hypersensitivity—i.e., are accessible [27,30]. Importantly, these HRE motifs are already open and accessible before the cells are made hypoxic and thus before HIF is stabilized [27,30]. Furthermore, hypoxia does not appreciably alter accessibility at HIF-binding sites. These findings suggest that, in general, HIF does not act as a pioneer factor (a transcription factor that binds to condensed chromatin and alters its accessibility), but rather operates on enhancers that are already defined, before it is activated. Further evidence of this comes from ChIP-seq analysis of histone H3K4me1 and H3K27ac modification. Both marks are already present at (promoter-distal) HIF-binding sites before HIF is stabilized [30]. The presence of H3K27ac as well as H3K4me1 in normoxia suggests that these enhancers are already active and likely already bound by other transcription factors. Whilst H3K4me1 is constitutive, H3K27ac increases concomitant with HIF binding, consistent with an increase in overall enhancer activity. Interestingly, HIF-1 binds preferentially to sites marked by H3K4me3, while HIF-2α binds preferentially to sites marked with H3K4me1 [28]. However, it is not known whether these associations help drive differences in HIF-1 and HIF-2 binding or whether they are simply a consequence of the differing distributions of each isoform in relation to the TSS (HIF-1 is observed to bind closer to gene promoters, whilst HIF-2 binding is more often distant from the TSS) [27,28].

Notwithstanding the above, analysis of ChIP-seq and DNase-seq data reveals that while accessibility is a prerequisite for HIF binding it is not sufficient and other factors also influence whether a given HRE is occupied [27]. One such factor is DNA methylation. Although this broadly correlates with accessibility (accessible regions of chromatin have low rates of methylation), methylation of the CpG at the heart of the HRE motif (RCGTG) has been shown to prevent HIF binding both in vitro and in vivo [31,32,33,34,35,36,37]. Importantly, DNA methylation can inhibit HIF binding at the EGLN3/PHD3 locus, which is partially responsible for hydroxylating HIF, thereby disrupting this negative feedback loop on HIF expression [36,37]. Furthermore, as will be discussed below, DNA methylation can also be influenced by hypoxia [38,39,40].

Similar to HIF-binding sites, the promoters of HIF-target genes are also accessible and marked with H3K4me3 in normoxia, prior to activation by HIF [29,41,42]. This is perhaps less surprising, since with a few notable exceptions (e.g., CA9—carbonic anhydrase 9), most HIF-target genes are already expressed in normoxia and simply display “enhanced” expression in hypoxia [41,43]. Furthermore, these genes already have RNApol2 at their promoters and across their gene bodies, suggesting active gene transcription [29,41,44]. However, the ratio between promoter-associated RNA-pol2 and that across the gene body falls in hypoxia when HIF is stimulated, indicating that HIF is acting as an RNApol2 pause-release factor. Indeed, HIF-1α has been shown to recruit the mediator-associated kinase (CDK8) and/or TIP60 to stimulate RNApol2 elongation [44,45].

To date, there have been relatively few analyses of higher-order chromatin structure and looping in relation to HIF-binding sites and the promoters of HIF-target genes. Nonetheless, the existing studies have confirmed long-distance interactions between HIF-binding sites and their respective target promoters [28,46]. Furthermore, like the histone modifications at HIF-binding sites and target gene promoters, these interactions are largely established in normoxia, prior to HIF stabilization. Hypoxia does not appear to substantially alter these interactions, although the number of sites examined remains relatively small.

Taken together, we propose a model for HIF-mediated transcriptional activation, in which enhancers and promoters at HIF-target genes are already epigenetically marked as such and are already interacting with each other, possibly through chromatin looping. Binding of HIF to this pre-established enhancer-promoter complex recruits elongation factors that then allow promoter-paused RNApol2 to be released. In turn, this causes gene transcription to increase rapidly in response to physiological or pathological hypoxia. 

Although the epigenetic environment clearly shapes the HIF response, only a proportion of hypoxia- and HIF-regulated genes bind HIF directly, indicating that other (less direct) mechanisms of gene regulation are also in operation [27]. Although HIF regulates a number of additional transcription factors and transcriptional repressors, some of which may also be directly regulated by hypoxia, much work has focused on epigenetic modifications as a potential additional mechanism to regulate gene expression in hypoxia. Such modifications may act to enhance or repress the activation of genes directly and/or to modify the response to the HIF-transcriptional pathway.

## 4. Oxygen-Sensitivity of 2-Oxoglutarate-Dependent Epigenetic Modifiers

Initial interest in the effects of oxygen availability on the epigenetic landscape stemmed from the realization that the family of 2-oxoglutarate-dependent dioxygenases (which act as oxygen-sensors in the HIF pathway) include a number of enzymes with roles in both histone and DNA modification. In particular, Jumonji C (JmjC) domain-containing 2-oxoglutarate-dependent dioxygenases comprise the largest group of histone lysine demethylases (KDMs) [22]. Each shares a common distorted double-stranded β-helix (DSBH, or “jelly-roll”) catalytic domain. JmjC KDM demethylation starts with hydroxylation to form a hemiaminal that subsequently fragments to form the demethylated lysine product and formaldehyde [22]. In addition to their catalytic domains, most JmjC KDMs have binding domains that assist in substrate specificity as well as recruitment to specific chromatin regions. This allows JmjC KDMs to be sub-classified into seven subfamilies (KDM2-8) that act on different lysine residues and methylation states. Broadly, KDM2s act on H3K36me1/2, KDM3s act on H3K9me1/2, KDM5s act on H3K4me2/3, KDM6s act on H3K27me2/3, KDM8 acts on H3K36me2/3, while KDM4s are more promiscuous acting on H3K9me2/3, H3K36me2/3, and H1.4K26me2/3 and KDM7s demethylate H3K9me1/2 and H3K27me1/2 [47,48] (Figure 2). In addition, some KDMs have also been shown to act on methylarginine residues, while others have unassigned activities and the role of JMJD6 is remains controversial.

As well as acting on proteins, 2-oxoglutarate dioxygenases can also take nucleic acids within DNA or RNA molecules as substrates. Most AlkB homologs, including FTO (fat-, mass, and obesity-associated protein), are 2-oxoglutarate-dependent dioxygenases that act as DNA/RNA demethylases involved in DNA repair or in controlling RNA stability [22]. The ten-eleven translocation (TET) family of 2-oxoglutarate-dependent dioxygenases catalyze sequential oxidation of methylcytosine (an important DNA modification in gene regulation) to 5-hydroxymethylcytosine (5hmC), 5-formylcytosine (5fC), and 5-carboxycytosine (5caC). Although important modifications in their own right, 5fC and 5caC can be removed by DNA glycosylases and base excision repair machinery to regenerate unmethylated cytosine. This provides a critical mechanism for erasing DNA-methylation with important consequences for the epigenetic regulation of gene expression [8].

However, in order to act as physiological oxygen-sensors, 2-oxoglutarate-dependent dioxygenases must have the ability to respond to changes in oxygen concentration within the range normally encountered within the cell. This has been reported for the HIF hydroxylases (Table 1), which have the highest in vitro Km values for oxygen of any 2-oxoglutarate-dependent dioxygenase [49,50,51,52]. This means that enzyme activity varies linearly with oxygen concentration and generates a graded response in HIF-α levels throughout the physiological range. Conversely, collagen prolyl hydroxylase (another 2-oxoglutarate-dependent dioxygenase) has a low Km for oxygen, such that it is fully active, even at low levels of oxygen and its activity is unaffected by modest levels of hypoxia [49]. Thus, whilst collagen prolyl hydroxylase requires oxygen for activity in the same way as the HIF hydroxylases, it is poorly suited to act as a physiological oxygen sensor. To date, detailed kinetic analyses of JmjC KDMs, with respect to oxygen, have been limited and it is therefore difficult to know which are capable of acting directly as physiological oxygen-sensors. Nevertheless, values obtained for KDM3A, KDM4A, B, C, and E, and KDM6A [53,54,55,56,57] are intermediate between those of collagen prolyl hydroxylase and PHD2, suggesting that they may act as oxygen-sensors, albeit potentially over a different range of oxygen concentrations to HIF. Conversely, Km values of KDM5A-D, KDM6B, and the TET enzymes for oxygen are extremely low [56,58,59], indicating that their activity will only be altered as cells approach complete anoxia and that they will remain constitutively active across the physiological range of oxygen levels.

In addition to oxygen, the 2-oxoglutarate dependent dioxygenases also use 2-oxoglutarate as a co-substrate rendering their activity potentially susceptible to the abundance of this metabolite. Furthermore, naturally occurring analogues such as succinate (a product of the reaction) and fumarate, may act as competitive inhibitors of this enzyme family. Indeed, the role of these metabolites in inhibiting the HIF hydroxylases in certain cancers is well established [60,61,62,63]. Conversely, mutations in IDH in certain brain cancers lead to the production of the oncometabolite, R-2-hydroxyglutarate, which potentiates the activity of the HIF hydroxylases [64]. The role of these metabolites in activating or inhibiting the histone demethylases has also been studied [65]. In vitro Km values for 2-oxoglutarate for KDM4A, KDM5B, KDM6A, and KDM6B varied from 6 to 50 μM, with KDM4B having the highest affinity and KDM6B the lowest. All the recombinant enzymes studied were susceptible to competitive inhibition by 2-oxoglutarate analogues to varying degrees and specificities. Similarly, succinate and fumarate have both been reported as potent inhibitors of the TET DNA methylase enzymes [58]. Thus, metabolic intermediates as well as oxygen have the capacity to alter both histone and DNA methylation by modulating activity of these enzymes. However, it appears that changes in these metabolites do not contribute to changes in histone modifications in hypoxia [56,66].

## 5. Oxygen-Dependent Changes in Enzyme Abundance

As well as requiring oxygen for enzyme activity, many 2-oxoglutarate-dependent dioxygenases are themselves transcriptionally regulated by hypoxia. In particular, the HIF prolyl hydroxylases, PHD2 and 3 are direct transcriptional targets of HIF [41]. This provides a negative feedback mechanism, whereby inhibition of HIF prolyl hydroxylase activity in hypoxia leads to induction of enzyme levels that act to abrogate HIF-α induction. Indeed, with prolonged hypoxia, HIF-levels fall, although remain above normoxic levels. Similarly, many of the JmjC KDMs and TET enzymes, as well as DNA methyltransferases (DNMTs), have been found to be direct transcriptional targets of HIF (Table 2) [29,40,67,68,69,70,71,72,73,74,75,76,77,78,79,80,81,82]. In each case, induction of KDMs by HIF will act to increase demethylase activity, thereby reducing methylation of the targeted mark. Importantly, these enzymes may also be targeted by other cancer-associated pathways (e.g., regulation of KDM4B by HIF and estrogen receptor in breast cancer cells) and this convergence emphasizes the importance of the induction of these enzymes in cancer [83,84]. However, as we have already seen, the enzyme activity of KDM3A, KDM4B, and KDM4C, which exhibit consistent induction by HIF, is inhibited by hypoxia [53,54,56]. Therefore, the overall effect on methylation at H3K9, H3K36, and H1.4K26 (a linker histone that binds DNA between nucleosomes to facilitate formation of chromatin fibers) will be a balance of effects. It is likely that the same holds true for many other HIF-inducible KDMs for which the Km for oxygen remains to be determined. 

Although the TET enzymes do not respond directly to oxygen across the physiological range [58,59], TET2 and TET3 can be hydroxylated by the same prolyl hydroxylase enzymes (PHD2/EGLN1 and PHD3/EGLN3) that hydroxylate HIF and can be targeted for proteasomal degradation by the same ubiquitin E3 ligase (VHL) in response to this hydroxylation [85]. This provides a mechanism by which DNA methylation may be reversed in hypoxia. Furthermore, both DNA methyltransferases (DNMTs) as well as TET enzymes may also be transcriptionally induced by HIF [40,77,78,79,80,81,82], such that the overall effect on 5mC levels will again be a balance of competing effects. Whilst studies suggest that this balance might favor hypomethylation in hypoxia, Skowronski et al. reported this in association with hypoxic suppression (rather than induction) of DNMT1, DNMT3A, and DNMT3B mRNA and protein, suggesting that effects may differ in different settings [38,39,40].

## 6. The Effect of Hypoxia on Global Histone Modification

The presence of competing effects on epigenetic modifications has led many groups to examine the direct effect of hypoxia on histone modifications. A range of methodologies have been employed, including immunoblotting of nuclear or histone extracts, mass-spectrometry, immunohistochemistry, and enzyme-linked immunosorbent assay, although the former is most prevalent (Table 3) [55,56,66,77,86,87,88,89,90,91,92,93,94,95,96,97,98,99,100,101]. Although hypoxic stimuli vary widely in both the severity (0.2–10% O_2_) and duration (30 min to 14 weeks) of hypoxia and in the cell-line used, consistent patterns are observed. However, given the heterogenous conditions used it is difficult to get an impression of the oxygen-sensitivity of each modification and/or the time course over which it manifests. Some changes appear transient or even biphasic, while others are only reported after prolonged hypoxia. Furthermore, there is clearly a publication bias in reporting these findings as very few groups have reported histone modifications that do not change under their conditions. This makes it even harder to get a full sense of the effects of severity and duration of hypoxia or whether hierarchical responses exist. Generally, methylation increases at histone H3K4. This includes both the H3K4me1 enhancer mark [90], but also particularly the H3K4me3 mark [29,66,90,91,95,97,99,101] seen at the promoters of highly transcribed genes. H3K36me3 (another mark that positively correlates with gene expression) is also increased in hypoxia [29,55,66,88,96]. However, both H3K9me3 and H3K27me3 (marks associated with repressed genes) are also consistently increased in hypoxia [55,56,66,87,90,91,92,93,94,95,96,102]. Thus, hypoxia-induced patterns of histone methylation are associated with both increased and reduced gene expression. In addition, the changes in histone modification observed on immunoblotting are often quite profound and must therefore reflect extensive changes across the genome rather than at just a few gene loci. Thus, it seems likely that global changes in histone modification in hypoxia will be associated with widespread, rather than locus-specific changes in gene expression. Unfortunately, the normalization inherent in pangenomic assays of gene expression can make detection of global changes in gene expression difficult to detect. 

Overall, hypoxia appears to increase histone methylation, consistent with the predominant effect arising from inhibition of enzyme activity rather than from transcriptional activation of JmjC KDMs by HIF. However, not all KDMs are HIF targets and so it is possible that the resultant effect of hypoxia on these specific KDMs is different from others. Furthermore, different KDMs (even those that share the same methylysine substrate) may be recruited to different regions of chromatin through their varied binding domains. Therefore, while methylation at H3K4, H3K9, H3K27, and H3K36 is increased globally, regions of chromatin specifically targeted by HIF-induced KDMs may remain unchanged or may even exhibit reduced methylation in hypoxia. 

Hypoxia also leads to a general decrease in histone acetylation [77,86,87,89,90,92,97,100] and can induce profound changes in cellular metabolism, including a reduction in acetyl-CoA, which histone acetyl-transferases require as a co-substrate in the acetylation of histones [92]. In hypoxic cells, restoration of acetyl-CoA levels can reverse de-acetylation, providing evidence for this direct mechanism [92]. However, specific lysine residues cannot be methylated and acetylated at the same time and so it is difficult to determine whether reciprocal changes in acetylation are a consequence of changes in methylation or occur directly.

Although HIF-bound enhancers and the promoters of HIF-target genes are generally accessible in normoxia [27,29,30,41,42], before HIF-binds, the observed global changes in histone methylation and acetylation raise the possibility of more widespread changes in chromatin accessibility. Indeed, Kirmes et al. showed that combined hypoxia and nutrient starvation led to a compacted chromatin state associated with resistance to digestion by DNase I [103]. Similarly, Li et al. observed resistance to DNase I in hypoxic cells, which was partially reversed by restoring acetyl-CoA dependent histone acetylation [92]. Furthermore, while Suzuki et al. observed no change in DNase I sensitivity at most loci in hypoxia, they did observe more subtle changes in nucleosome positioning at some loci using micrococcal nuclease (MNase) digestion [42]. However, using ATAC-seq, Ward et al. did not detect any changes in chromatin accessibility in hypoxia [104], while Miar et al. observed variable effects of hypoxia on DNA accessibility at different sites [105], and Wang et al. observed an overall increase in chromatin accessibility in hypoxic cells [106]. Thus, it remains unclear whether changes in histone modification in hypoxia correlate with changes in chromatin accessibility.

## 7. Locus-Specific Changes in Chromatin

Given the possibility that histone modifications might be altered in a locus-specific manner in hypoxia and the importance that this would have for specific gene regulation, many groups have examined for such changes using chromatin immunoprecipitation (ChIP) coupled to quantitative PCR. Whilst these studies are again heterogenous in the cell-lines studied, the duration and severity of hypoxia used, and the gene loci examined (Table 4) [41,42,56,66,87,88,89,90,91,93,96,97,107,108,109,110], several patterns can be distinguished. Firstly, with the exception of very severe (0.01%) hypoxia, methylation of H3K4 is either increased or unchanged at the genes examined. Similarly, with the exception of H3K9me2 at the EGR1 and VEGF loci in Hepa 1-6 cells and H3K9me3 at the CC2D2A and HSD17B4 loci in hADSC cells, methylation is generally increased at H3K9 as well. However, hypoxia-induced changes in H3K27me3 appear to vary much more between loci. Although histone acetylation is examined in relatively few studies, H3K4ac is consistently reduced in hypoxia, and H3K9ac increases or decreases at different gene loci, whilst H3K27ac is consistently induced.

Whilst the small number of loci reported in each study makes it difficult to draw firm conclusions, taken together, these results indicate a degree of locus-specificity for changes in histone modifications in hypoxia. This has led several of the groups to examine pangenomic changes in histone modification by coupling chromatin immunoprecipitation to high-throughput sequencing (ChIP-seq) [30,41,56,66,110]. In these studies, the lack of spike-in controls and the use of conventional normalization strategies will negate many of the global changes observed in the immunoblotting analyses making absolute changes difficult to interpret, and this should be addressed in any future work. Furthermore, there may be important differences in the oxygen-sensitivity and/or the time course of changes in individual modifications and at specific loci. This would have important biological consequences and may provide greater mechanistic understanding. This important question could be addressed in future analyses using high-throughput sequencing technologies. Nevertheless, Chakraborty et al. reported hypoxic induction of the repressive H3K27me3 mark at several genes (Actc1, Myl1, and Myog) that were repressed in hypoxia [56]. Conversely, Lee et al. did not observe changes in H3K27me3 at differentially expressed genes [91]. Instead, they described four groups of differentially expressed genes, based on changes in single trimethylation marks. One set of hypoxia-inducible genes showed induction of promoter-associated H3K4me3 alone. The other three groups of gene loci were identified based upon changes in H3K9me3. Two of these groups exhibited an inverse correlation between hypoxic changes in gene expression and changes in the repressive H3K9me3 mark across the gene bodies, as might be expected. However, they also reported a fourth set of hypoxia-inducible genes, which exhibited induction of promoter-associated H3K9me3. Batie et al. showed a positive correlation between hypoxic changes in H3K36me3 at gene loci and hypoxic-regulation of gene expression [66]. Similar to Lee et al., they also observed an increase in promoter-associated H3K4me3 signal at a subset of hypoxia-inducible genes, which they identified to be direct transcriptional targets of HIF. Likewise, Choudhry et al. observed hypoxic induction of H3K4me3 (and H3K27ac) signal that was limited to the promoters of HIF-target genes [41]. Further developments in technology such as CUT&RUN [111,112] or CUT&Tag [113] approaches are enabling increasingly higher resolution and lower background analyses from lower cell numbers and even from single cells, which may improve future understanding of cell-to-cell variability in the hypoxic response.

While it is tempting to ascribe changes in gene expression to hypoxia-associated changes in histone modification, causality cannot be assumed from simple associations. This is epitomized by the behavior of HIF target genes in hypoxia, which exhibit increased promoter-associated H3K4me3 (and H3K27ac) and increased expression, although these genes are activated by HIF rather than by hypoxia-dependent histone modifications per se. Indeed, both HIF itself and the RNApol2 complex that it activates are known to recruit both histone methyltransferases and histone acetyl transferases, making it possible that HIF-mediated changes in gene expression might lead to changes in epigenetic modification.

## 8. Recruitment of Histone Modifying Activity by HIF

Perhaps the best described epigenetic regulators associated with the HIF transcription factor are the p300 and CBP (CREB-binding protein) transcriptional co-activators. These closely related proteins both bind to the C-terminal transactivation domain of HIF-α to enhance transactivating ability [114,115]. This interaction is regulated by oxygen-dependent asparaginyl hydroxylation of HIF-α, mediated by the 2-oxoglutarate-dependent dioxygenase, FIH [24,25], as well as by HIF abundance. In addition to a number of protein-binding domains, p300 and CBP also have domains pertaining to histone binding and modification. They each have a bromodomain that recognizes acetylated lysines, as well as a histone acetyltransferase domain, and are writers of the H3K27ac mark [116,117]. Therefore, binding of HIF would be expected to increase H3K27ac as observed at both HIF-binding sites and at the promoters of HIF-target genes [30,41]. In addition, HIF can also interact with TIP60, the CDK8-mediator complex, or both, as alternate co-activators for gene expression [44,45]. Although CDK8 mediates RNApol2 elongation without affecting histone acetylation, TIP60 has inherent histone acetylase activity and plays an important role in H3K9 acetylation at HIF target genes [45]. Alternately, HIF-1 can interact with other co-activators, SRC-1 and SRC-3 (NCOA1/3), which also have histone acetyltransferase activity [118,119,120,121].

In addition to acetyltransferases, HIF also interacts with both writers and erasers of histone methylation. HIF-1 can interact with the histone demethylase, KDM3A, which is recruited to the SLC2A3 locus, where it demethylates the repressive H3K9me2 mark to help upregulate its expression in hypoxia [122]. Similarly, HIF-1, but not HIF-2 can interact with the KDM4C histone demethylase, which demethylates H3K9me3 at the BNIP3, LDHA, PDK1, and SLC2A1 HIF-target gene loci, contributing to gene induction [123]. Conversely, the H3K4 methyltransferase SET1B has recently been shown to bind HIF-1, contributing to both the induction of H3K4me3 at the promoters of HIF target genes and to gene induction [James Nathan, personal communication]. Finally, TET1 also binds to both HIF-1 and HIF-2 to enhance their transactivation activity, although this is thought to be independent of its ability to oxidize methyl cytosine [80]. Thus, hypoxic regulation of epigenetic modifications operates not only at the level of enzyme abundance and enzyme activity, but also at the level of recruitment to chromatin to generate locus-specific effects.

## 9. Post-Translational Modification of HIF by Epigenetic Modifiers

In addition to recruiting epigenetic modifiers to act on chromatin, HIF-α is also directly methylated or acetylated itself by a number of these enzymes, and this may have important consequences for both its stability and its transcriptional activity (Figure 3). Specifically, p300 acetylates HIF1a Lys709, which stabilizes the protein in both normoxia and hypoxia [124] and can be reversed by the sirtuin 2 (SIRT2) deacetylase [125]. Similarly, p300/CBP associated factor (PCAF) is also able to acetylate HIF-1α at Lys674, again stabilizing the protein. This can, in turn, be reversed by the sirtuin 1 (SIRT1) deacetylase [126,127]. Conversely, ARD1-mediated acetylation of Lys532 may facilitate interaction with VHL to de-stabilize HIF-1α [128], whilst the histone deacetylases HDAC1, 2, 3, 4, and 6, can deacetylate HIF-1α and increase its stability [129,130,131,132]. In addition, HIF-α may be methylated by G9a [87,133], SET7 [134], or SET9 [135], with effects on both stability and transcriptional activity.

Furthermore, epigenetic modifiers may cause locus-specific changes to the HIF-1α and HIF-2α gene loci themselves, thus affecting their expression. In particular, KDM4A can remove the repressive H3K9me3 mark at the HIF-1α locus, enhancing its transcription [88]. Conversely, DNA methylation at the HIF-2α gene locus (EPAS1) by the DNA methtyltransferase, DNMT3A, silences HIF-2α expression [136]. Thus, epigenetic modifying enzymes may have effects on HIF-mediated transcription, either by altering HIF-1α protein itself or by modifying the HIF-1α or HIF-2α gene loci.

## 10. Changes in Histone Modification as a Consequence of Gene Activation

As well as directly recruiting epigenetic modifiers, HIF also interacts with the core transcriptional machinery, including RNApol2 and many general transcription factors (GTFs) to regulate transcription in a series of ordered steps known as the transcription cycle [116,137]. (1) Formation of the pre-initiation complex (PIC) is facilitated by reader proteins, including TAF3, that bind to promoter-associated H3K4me3. Many, but not all, genes are regulated through transcription initiation. (2) The transition between initiation and elongation through release of paused RNApol2 is another key step in transcription, which is regulated at 30–70% of metazoan genes, including HIF target genes [41,44,45]. This is affected through positive transcription elongation factor b (P-TEFb)-mediated phosphorylation of RNApol2 and is facilitated by the binding of BRD4 or YEATS domain-containing proteins to acetylated histones. (3) Once RNApol2 is released, it undergoes productive elongation assisted by a number of additional protein complexes. In particular, phosphorylated RNApol2 recruits the SET1A, SET1B, and MLL H3K4me3 methyltransferases, thereby increasing this mark downstream of the transcriptional start site [138,139] as observed at HIF-target genes [41]. SETD2 travels with elongating RNApol2 whilst depositing H3K36me3. In turn, this recruits the Rpd3S histone deacetylase complex, the H3K4me3 demethylase, KDM5B, and the DNA methyltransferases, DNMT3a and DNMT3b, which increase DNA methylation across the gene body. Similarly, DOT1L also travels with elongating RNApol2 whilst depositing methyl marks on H3K79. (4) Finally, transcription termination occurs with eviction of RNApol2 from DNA, which involves G9a mediated deposition of H3K9me2. Thus, not only do histone modifications affect operation of the core transcriptional machinery, but the act of transcription also alters the local epigenetic environment in an interactive manner in which the two, bidirectional processes are inextricably linked.

## 11. Epigenetic “Memory”

While hypoxia induces changes in chromatin, it is not clear how long these changes persist following reoxygenation, and whether they can affect the transcriptional response to subsequent episodes of repeated hypoxia and thereby contribute to the phenomenon of hypoxic-preconditioning [140]. Indeed, Prickaerts et al. showed that global H3K4me3 and H3K27me3 fell rapidly upon reoxygenation [95], Olcina et al. saw a rapid reversal of H3K9me3 [101], and Batie et al. observed a rapid return to basal levels for global H3K4me3, H3K9me3, H3K27me3, H3K36me2, and H3K36me3 within 1 h of reoxygenation [66]. Furthermore, other groups have shown rapid reversal of hypoxia-induced H3K4me3, H3K9me3, H3K27me3, and H3K9ac at specific gene loci [42,93,95,110]. This would suggest that hypoxia-induced changes in epigenetic modifications are not maintained upon reoxygenation. However, it is possible that transcriptionally induced changes in the abundance of epigenetic modifiers may persist for a more prolonged period following re-oxygenation and contribute in some way to a lasting epigenetic effect. In this respect, Zhang et al. noted a persistent fall in DNMT3A and DNMT3B mRNA in the hippocampus of preconditioned mice [141]. Therefore, the question of whether hypoxia generates lasting changes in chromatin in the normal physiological setting remains an open question that could be addressed in future studies.

## 12. Discussion

While much of the evidence described above is from cultured cancer and/or immortalized cell lines, which may differ from normal physiological conditions, the association between the epigenetic environment and transcriptional regulation in hypoxia is clearly highly complex (Figure 4). Much of the landscape in which HIF operates is set by DNA-accessibility, CpG methylation, histone modification, and chromatin looping that is already present in normoxia before HIF is stabilized. However, hypoxia can further modify both the global and locus-specific chromatin landscape; either directly by influencing enzyme activity or indirectly through HIF-mediated transcriptional activation of epigenetic modifiers. This may both extend the range of genes regulated by hypoxia as well as amplifying the magnitude of the response. Furthermore, epigenetic modifiers may act on gene loci within the HIF pathway or may directly modify the proteins themselves, thereby influencing HIF transcriptional activity in a feedback loop. In addition, epigenetic modifiers are an integral part of the transcriptional machinery and are recruited to specific loci, both by HIF itself and by the basal transcriptional complex. Thus, epigenetic modifications are both consequent on and contribute to the regulation of transcription in hypoxia. Therefore, when considering the relationship between transcriptional and epigenetic regulation in hypoxia, it is not so much a question of cause or effect, but rather a matter of both cause and effect with the degree and extent to which they are interlinked remaining to be fully determined.

## Figures and Tables

**Figure 1 ijms-21-08320-f001:**
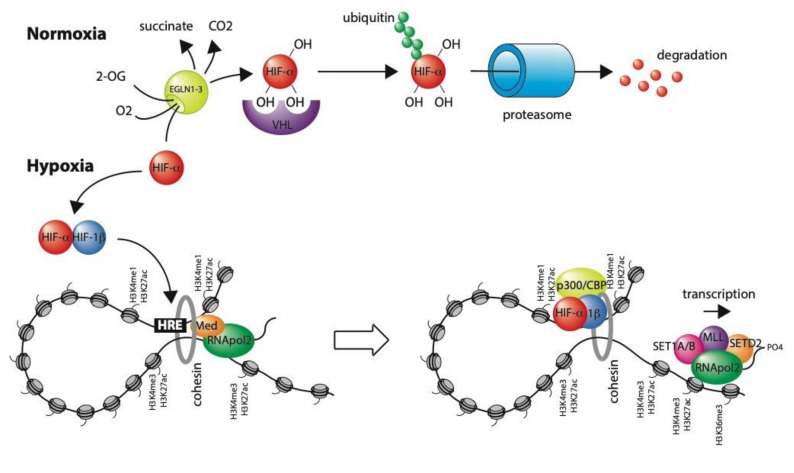
Overview of the HIF transcription factor in normoxia and hypoxia. In normal oxygen conditions (normoxia), the oxygen-dependent prolyl hydroxylases (EGLN1-3) hydroxylate HIF-α. This hydroxylation allows the von Hippel-Lindau E3 ubiquitin ligase to bind HIF-α and to covalently add ubiquitin moieties that target it for proteasomal degradation. In low levels of oxygen (hypoxia), the activity of the EGLN enzymes is reduced and HIF-α is stabilized. It then forms a heterodimer with HIF-1β and translocates to the nucleus, where it binds to hypoxia response elements (HREs) to release promoter-paused RNApol2 and enhance gene transcription. Often, these HREs are in distant enhancer regions, which contact their target promoters through cohesin-mediated chromatin looping. In normoxia, before HIF is stabilized, HIF-binding sites and target promoters are generally accessible and display histone modifications associated with active enhancers and promoters. However, both HIF and activated RNApol2 recruit additional essential epigenetic modifying activities that further modify the chromatin as a result of HIF-mediated transactivation.

**Figure 2 ijms-21-08320-f002:**
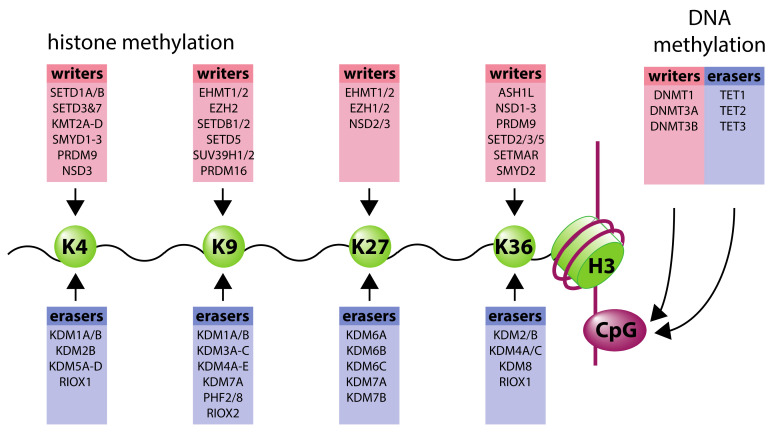
Writers and erasers of histone H3 methyl lysine, and DNA CpG methylation.

**Figure 3 ijms-21-08320-f003:**
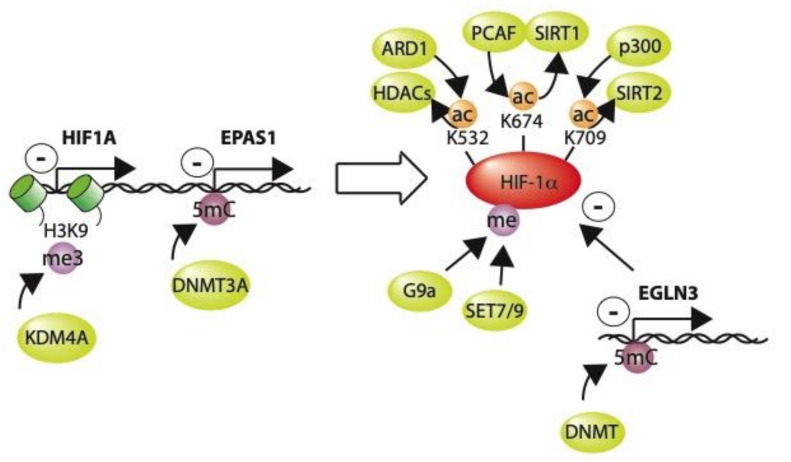
Histone and DNA modifying enzymes that affect HIF levels. Removal of the repressive H3K9me3 mark by KDM4A at the HIF1A locus activates transcription of HIF-1α. Conversely, CpG methylation at the EPAS locus suppresses transcription of the HIF-2α isoform. Reversible acetylation of multiple residues on HIF-1α has variable effects on HIF-1α stability. Similarly, methylation of the molecule by G9a or SET7/9 also affects its stability. Methylation of the EGLN3 promoter suppresses transcription of this negative regulator of HIF.

**Figure 4 ijms-21-08320-f004:**
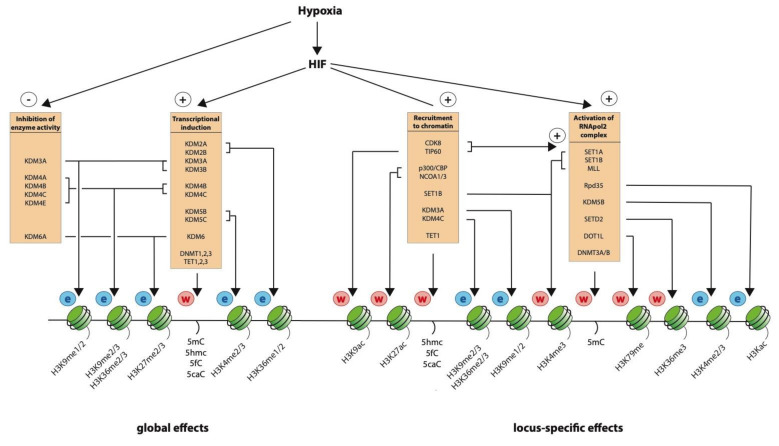
Regulation of chromatin structure by hypoxia on a global and locus-specific level. Independent of the HIF transcriptional pathway, hypoxia globally inhibits the activity of many 2-OG dependent dioxygenases that control histone and DNA modifications. The HIF transcription factor complex, activated by hypoxia, also works to control histone and DNA modifications on a global level by transcriptional induction of these same readers and writers. To cause locus-specific effects, HIF recruits these enzymes to chromatin and activates the RNApol2 complex.

**Table 1 ijms-21-08320-t001:** Sensitivity of 2-oxoglutarate-dependent dioxygenases to oxygen. 2-OG dependent dioxygenases that have been shown to have histone and DNA modifying activities, the targets they modify, and their sensitivity to changes in oxygen concentration.

Sensitivity of 2-OG-Dependent Dioxygenases to Oxygen			
Enzyme	Reference	Km for Oxygen (mM)	Target
PHD2	Hirsila, 2003 [49]	250	HIF
	Ehrismann, 2007 [50]	250	
	Dao, 2009 [51]	1746 ± 574	
	Tarhonskaya, 2014 [52]	> 450	
FIH	Ehrismann, 2007 [50]	90-237	
C-P4H	Hirsila, 2003 [49]	40	collagen
KDM3A	Qian, 2019 [53]	7.59% ± 0.80%	H3K9me1/2
KDM4A	Cascella, 2012 [54]	57 ± 10	H3K9me2/3, H3K36me2/3, H1.4K26me2/3
	Hancock, 2017 [55]	173 ± 23	
	Chakraborty, 2019 [56]	60 ± 20	
KDM4B	Chakraborty, 2019 [56]	150 ± 40	
KDM4C	Cascella, 2012 [54]	158 ± 13	
KDM4E	Cascella, 2012 [16]	197 ± 16	H3K9me2/3
	Sanchez-Fernandez, 2013 [57]	> 93	
KDM5A	Chakraborty, 2019 [56]	90 ± 30	H3K4me2/3
KDM5B	Chakraborty, 2019 [56]	40 ± 10	
KDM5C	Chakraborty, 2019 [56]	35 ± 10	
KDM5D	Chakraborty, 2019 [56]	25 ±5	
KDM6A	Chakraborty, 2019 [56]	200 ± 50	H3K27me3
KDM6B	Chakraborty, 2019 [56]	25 ± 5	
TET1	Laukka, 2016 [58]	30	methylcytosine
	Thienpont, 2016 [59]	0.31%	
TET2	Laukka, 2016 [58]	30	
	Thienpont, 2016 [59]	0.53%	

**Table 2 ijms-21-08320-t002:** Chromatin modifying enzymes that are direct transcriptional targets of HIF. Histone and DNA modifying enzymes that are directly regulated by the HIF transcription factors and the specific modification they add or remove.

Chromatin Modifying Enzymes Targeted by HIF																		Target
KDM2A										✔								H3K36me1/2
KDM2B										✔								
KDM3A	✔	✔	✔	✔	✔	✔												H3K9me1/2
KDM3B					✔													
KDM4B	✔	✔		✔	✔	✔	✔											H3K9me2/3, H3K36me2/3, H1.4K26me2/3
KDM4C	✔			✔	✔													
KDM5B				✔		✔												H3K4me2/3
KDM5C								✔										
KDM6B									✔									H3K27me2/3
JMJD6					✔													
PLU-1					✔													
SMCX					✔													
RBP2					✔													
KIAA1718					✔													
TET1														✔	✔	✔	✔	methylcytosine
TET2																	✔	
TET3																✔	✔	
DNMT1											✔	✔	✔					cytosine
DNMT3A											✔							
DNMT3B											-	✔						
	Pollard, 2008 [67]	Beyer, 2008 [68]	Wellman, 2009 [69]	Xia, 2009 [29]	Yang, 2009 [70]	Krieg, 2010 [71]	Fu, 2012 [72]	Niu, 2012 [73]	Lee, 2014 [74]	Batie, 2017 [76]	Liu, 2011 [40]	Watson, 2014 [77]	Xu, 2018 [78]	Mariana, 2014 [79]	Tsai, 2014 [80]	Wu, 2015 [81]	Lin, 2017 [82]	

**Table 3 ijms-21-08320-t003:** Global changes in histone modifications in hypoxia. Publications showing global changes in histone methylation and acetylation in hypoxia across various cell types, oxygen levels, and duration of hypoxia. Up arrows indicate global induction of specific PTMs, down arrows indicate global reduction, and dashes indicate no significant change. ac indicates acetylation at that specific histone subunit and lysine residue, while me1, me2, and me3 indicate mono-, di-, and tri- methylation, respectively.

Global Changes to Histone Modifications in Hypoxia																								
Reference	Cell Line(s)	% O_2_	Time (h)	H2AK5	H3K4				H3K9				H3K14	H3K16	H3K27			H3K36		H3K79	H4	H4K5	H4K12	H4R3
				ac	ac	me1	me2	me3	ac	me1	me2	me3	ac	ac	ac	me2	me3	me2	me3	me2	ac	ac	ac	me2
Costa, 2005 [86]	A549	0.5	1.5–9						↓	↓	↑	↑									↓			
Chen, 2006 [87]	A549, HOS, HEK293, MES	0.5	1.5–24						↓	↓	↑	↑												
Islam, 2006 [89]	Fetal lung type II	2	24						↓		↑													
Johnson, 2008 [90]	Hepa 1-6	0.2	48			↑	↑	↑	↓		↑		↑			↑	↑			↑				↑
Xia, 2009 [29]	HepG2	0.5–5	24				↑	↑			↑								↑					
Zhou, 2010 [99]	Beas-2B, A549	1	6–48					↑																
Tausendschon, 2011 [96]	RAW254.7	1–8	24								↑	↑							↑					
Wu, 2011 [97]	FADU, MCF-7	1	18	↓	↓		↑	↑				-					-					↓	↓	
Olcina, 2013 [101]	RKO	<0.1, 2	6 to 18								↑	↑					-							
Watson, 2014 [77]	PwR-1E	10% × 7wks, 3% × 4wks, then 1% × 3wks							↓															
Osumek, 2014 [94]	McA-RH777	1, 5	24 or 48									↑												
Dmitriev, 2015 [100]	PC12	0 & no glucose	1 to 9											↓										
Olcina, 2016 [93]	RKO	<0.1, 2	6–48									↑												
Prickaerts, 2016 [95]	MCF-7	<0.2	8 or 24					↑									↑							
Dobrynin, 2017 [88]	RKO	0.1, 2	24								↑								↑					
Hancock, 2017 [55]	U2OS	0.1–5	24					-				↑					↑		↑					
Lee, 2017 [91]	hADSC	<0.5, 1, 2	24 or 48					↑				↑					↑							
Batie, 2019 [66]	HeLa, HFF	1	0.5–24					↑				↑					↑	↑	↑					
Chakraborty, 2019 [56]	mHepa-1 c4	5	96									↑					↑							
Li, 2020 [92]	CHP134,SMS- KCNR, MEF	0.5	6 or 24						↓			↑			↓		↑							

**Table 4 ijms-21-08320-t004:** Locus-specific changes in histone modifications in hypoxia. Publications showing locus-specific changes in histone methylation in hypoxia, as well as any associated change in expression of that gene, in a wide range of cell types, oxygen levels, and duration of hypoxia. Up arrows indicate induction specific PTMs, down arrows indicate global reduction, and dashes indicate no significant change. ac indicates acetylation at that specific histone subunit and lysine residue, while me1, me2, and me3 indicate mono-, di-, and tri- methylation, respectively.

Locus-Specific Changes to Histone Modifications in Hypoxia
Reference	Cell Line(s)	% O_2_	Time (h)	Gene Locus	H3K4	H3K9	H3K27	Expression
ac	me1	me2	me3	ac	me2	me3	ac	me3
Islam, 2006 [89]	Human fetal	2	24	SP-A						↑				↑
Chen, 2006 [87]	A549	0.5	6	Dhfr, Mlh1						↑				↓
			Cap43						-				
Johnson, 2008 [90]	Hepa 1-6	0.2	48	AFP, ALB				↑	↓	↑			↓	↓
			EGR1, VEGF				↑	↑	↓			↓	↑
			Brn3-b				-		↑			↓	
Lu, 2011 [107]	MCF-7 or RKO	0.01	12–72	RAD51, BRCA1			↓	↓	↓		↑			↓
			VEGF			↑	↑	↑		↑			↑
Wu, 2011 [97]	FADU	1	18	CDH2, VIM	↓		↑						↓	↑
			CDH1	↓		↑			↑	↑		↑	↓
			JUP	↓		↑						↑	↓
MCF-7	1	18	CDH1						↑	↑			↓
Tausendschon, 2011 [96]	RAW254.7	1	24	Ccl2, Ccr1, Ccr5						↑	↑			↓
			ADM						-	-			↑
Choudhry, 2014 [41]	MCF-7	1	24	ALDOA, ADM				↑						↑
Lu, 2014 [108]	MCF-7	0.01	12–72	MLH1			↓	↓	↓		↑			↓
Schorg, 2015 [109]	MCF-7	0.5	16	PAG1		↑		-				↑		↑
			EGLN3								↑		
786-0	0.5	16	PAG1		↑		↑				↑		-
			EGLN3								↑		
Adriaens, 2016 & Prickaerts, 2016 [95,110]	MCF-7	<0.2	8 or 24	CCNA2, DPM1, NOL11, ATP2A3, FOXF1, IGFBP4				↑						
			ATF3, LPO, APLN, CYP1B1, SLC9A5									↑	
			GPRC5B, OPRL1				↑					↑	
			LOX				↑					↓	
Olcina, 2016 [93]	RKO	<0.1	6	APAK							↑			↓
Dobrynin, 2017 [88]	RKO	<0.1	24	HIF-1A							↑			↓
Lee, 2017 [91]	hADSC	<0.5	48	SLC22A15, PFKP, MEF2D, RUSC2							↑			↑
			PDE4C, PFKFB4, MT3, STC1				↑						↑
		SEC22B, BZW2, HNRNPA3, LUM							↑			↓
			CC2D2A, HSD17B4							↓			↓
Suzuki, 2018 [42]	SK-N-BE(2)c	1	4 or 24	CA9, PGK1					↑					↑
Batie, 2019 [66]	HeLa	1	1–24	BNIP3L, KLF10, LOX, ENO1, STAG2, CA9				↑						↑
			BAP1, KDM2B				-						-
			ACTB				↑						
Chakraborty, 2019 [56]	C2C12	2	96	Actc1, Myl1, Myog, Myh1, Myom3, Igfn1, Mb									↑	
			Adora1, Gjd2									-

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
