# Peer review of "Hypoxic Regulation of Gene Transcription and Chromatin: Cause and Effect"

_ijms, 2020, doi:10.3390/ijms21218320_

Round 1
Reviewer 1 Report
This is an extensive and thorough review of hypoxia/HIF from a perspective epigenetics. This reviewer really enjoyed reading this manuscript. While this reviewer acknowledged that this manuscript has touched many aspects of HIF/hypoxia functions in epigenetic regulation and vice versa, the authors may consider to discuss more about the impact of metabolism under hypoxia on epigenetics, such as 2-OG, fumarate and succinate as well as the oncometabolite 2-hydroxyglutarate.
In addition, hormone (estrogen) is also involved in regulation of HIF-1a and KDM4B, which may have an important effect on reproductive development and breast cancer.
At last, the author may discuss that all data so far were generated from cancer cell lines or immortalized cell lines, which may differ from normal physiological conditions.
Author Response
This is an extensive and thorough review of hypoxia/HIF from a perspective epigenetics. This reviewer really enjoyed reading this manuscript. While this reviewer acknowledged that this manuscript has touched many aspects of HIF/hypoxia functions in epigenetic regulation and vice versa, the authors may consider to discuss more about the impact of metabolism under hypoxia on epigenetics, such as 2-OG, fumarate and succinate as well as the oncometabolite 2-hydroxyglutarate.
We thank the reviewer for highlighting this important aspect. We have added a paragraph (lines 325-341) in the revised manuscript to address this.
In addition, hormone (estrogen) is also involved in regulation of HIF-1a and KDM4B, which may have an important effect on reproductive development and breast cancer.
We agree with the reviewer that the convergence of HIF and other cancer-associated pathways on histone modifiers emphasizes the importance of induction of these enzymes in cancer and have made reference to this using estrogen receptor and HIF-mediated regulation of KDM4B as an example (lines 363-366).
At last, the author may discuss that all data so far were generated from cancer cell lines or immortalized cell lines, which may differ from normal physiological conditions.
The reviewer is correct to point out that much of the evidence linking hypoxia, gene transcription and chromatin has been obtained from cultured cancer and/or immortalized cell lines and may not fully reflect normal physiology. We have amended the final discussion (lines 665-666) to highlight this important caveat.
Reviewer 2 Report
The manuscript “Hypoxic regulation of gene transcription and chromatin: cause and effect.” By Kindrick and Mole describes a review of the latest findings in hypoxia-induced factor (HIF) research with a focus on epigenetic modifications. It is a very detail oriented review that will serve new students to the subject well. What is missing are which open question remain to answered. How new technologies might help answer certain open questions. Are there competing models that need to be teased out? Overall, it is a timely review of how chromatin structure can be primed and subsequently further modified under stress conditions, hypoxia in this case. But besides the global point, several concerns remain and are mentioned below.
Concerns:
+ Throughout the manuscript HIF, HIF-1, HIF-2, HIF-1a and HIF-a are used seemingly randomly. This is confusing. Although HIF1 and HIF2 bind HREs, in mice they were found to play non-redundant roles. This does not become clear when reading this review. Please be consistent in precisely describing which HIF gene is studied when.
+ In the discussion the authors end with remarking that transcriptional and epigenetic regulation in hypoxia is a matter of both cause and effect. Yet, the authors have not discussed whether a hierarchical response exist that temporal to the extend and duration of hypoxic conditions. Overall, it is unclear how the authors come to the conclusion in the final sentence. Is there a set of experiments that could test their hypothesis?
+ It would help the reader if at the end of the introduction a brief statement is made what this review specially will address.
+ Page 3, line 128 – the authors mention that on average 4 enhancers interact with a promoter through cohesion-mediated chromatin loops. This reviewer is not sure if this number is as established as the authors describe. In fact, the existence of enhancer-promoter loops as means to facilitate their interaction is not always observed. For instance, recently the Bickmore’s group (Benabdallah et al 2010 Mol Cell) recently showed that upon activation the enhancer and promoter were further separated, contradicting the chromatin loop formation for promoter-enhancer interaction.
+ Page 4, line 130 – a recent paper by the Wysocka lab (Gu et al 2020 Mol Cell) very neatly describe how cohesin and transcription play opposing roles in CTCF cluster formation. It would be a nice addition if these recent results were briefly discussed in this paragraph.
+ The first paragraph on Page 5 reads a little difficult. For instance, it is unclear what “this” on line 175 refers to. How does HIF bind to accessible HREs and DHS’s? Doesn’t a DHS mean it is accessible? Studies Schodel et al and Platt et al are given a lot of weight as if they present fully established dogma. The term “pioneer factor” is not introduced. Please, improve the clarity of this important paragraph. Maybe, the authors can suggest future experiments with CUT&Tag and/or single cell ChIP-seq to better understand cell-to-cell variability in hypoxic response.
+ The second paragraph of Page 5 leaves the reader hanging how hypoxia can influence methylation.
+ The authors state that chromatin looping occurs (line 224), whereas in the preceding paragraph it is noted that very few higher-order chromatin structure studies have been done thus far. It might be more appropriate if the current state of affairs if summarized and the opinion or model of the authors is subsequently stated.
+ Page 6, line 240 – “the effects of oxygen and hypoxia” reads a bit weird. Oxygen is a molecule, whereas hypoxia is a biological state. Maybe the sentence can be rewritten to reflect the differences.
+ Page 7, line 276-7 – It is noted that 2-oxogluterate-dependent dioxygenase has a linear enzymatic response to O2 concentrations, it is not noted what the biological relevance of this observation is. This is however noted for prolyl hydroxylase (lines 279-81). Maybe this can be added.
+ Page 8, line 311 – The linker histone H1.4 is introduced without explanation what a linker histone is or is thought to do.
+ Table 2 – behind each v-mark, there appears to be an empty square. Is this supposed to be there? The font is also very small and difficult to read, especially in dim light conditions.
+ In section 6 a slew of chromatin studies are mentioned, but to this reviewer, what I am missing is a sense of how quick these changes were observed when cells were subjected to hypoxia conditions (even though they are reported in Table 4). In the hormone-gene expression field, it is widely expected that there are fast transcriptional responses directly induced by hormone receptors binding to genomic sites and subsequent medium and slow responsive genes. It would be informative if such data, as far as known, was clarified, as this would provide a mechanistic understanding in which order what event might occur.
+ Page 12, line 416 – “2” should be “Two”.
+ Page 12, last paragraph – it is confusing to this reviewer how at first the authors caution for overinterpretation of linking hypoxia-associated changes in histone modifications and gene expression, to end the paragraph with a HIF and RNAP2 to directly alter the histone modification landscape. Maybe it worth considering rewriting this paragraph to more clearly reflect the thought and concerns.
+ Section 9 ends rather abruptly. Maybe a concluding sentence/paragraph could be helpful for the reader.
+ Figure 3 – to this reviewer it is puzzling to see histone marks being added and removed from a cartoon of DNA that depleted of nucleosomes. Also, the variable effects of acetylation on HIF1a stability does not shine through. The figure suggests that K532, K674, and K709 first have to be acetylated and right away deacetylated, as well as methylated for it to be ultimately degraded. It is not clear that EGLN3 is a negative regulator of HIF1a.
+ In section 11, the authors discuss the potential of epigenetic “memory”. This paragraph is confusing how different findings are brought up and under which premise they are presented. Are HIF-responsive genes supposed to respond like cell-type specific marks? Are cultured cell lines directly comparable to mice studies? It might be worth rewriting this paragraph to clearly reflect what was found in which system and under which conditions, and how these findings differ or agree.
Author Response
The manuscript “Hypoxic regulation of gene transcription and chromatin: cause and effect.” By Kindrick and Mole describes a review of the latest findings in hypoxia-induced factor (HIF) research with a focus on epigenetic modifications. It is a very detail-oriented review that will serve new students to the subject well. What is missing are which open question remain to answered. How new technologies might help answer certain open questions. Are there competing models that need to be teased out? Overall, it is a timely review of how chromatin structure can be primed and subsequently further modified under stress conditions, hypoxia in this case. But besides the global point, several concerns remain and are mentioned below.
Thank you. We agree with the reviewer that it is important to draw the reader’s attention to open questions, for which we do not currently have answers. We have attempted to highlight these at various points in the manuscript, particularly in relation to the reviewer’s other comments as below.
Concerns:
+ Throughout the manuscript HIF, HIF-1, HIF-2, HIF-1a and HIF-a are used seemingly randomly. This is confusing. Although HIF1 and HIF2 bind HREs, in mice they were found to play non-redundant roles. This does not become clear when reading this review. Please be consistent in precisely describing which HIF gene is studied when.
Thank you. We use the term HIF-1alpha to refer to this subunit and HIF-1 to refer to the HIF-1alpha/HIF-1beta heterodimer and similarly, for HIF-2alpha and HIF-2. HIF-alpha refers generically to either HIF-1alpha or HIF-2alpha subunit whilst HIF refers to either HIF-1 or HIF-2 heterodimer. The reviewer correctly highlights the non-redundant roles of HIF-1 and HIF-2. We have therefore amended and added to lines 148-153 to clarify both of these points.
+ In the discussion the authors end with remarking that transcriptional and epigenetic regulation in hypoxia is a matter of both cause and effect. Yet, the authors have not discussed whether a hierarchical response exist that temporal to the extend and duration of hypoxic conditions. Overall, it is unclear how the authors come to the conclusion in the final sentence. Is there a set of experiments that could test their hypothesis?
This phrase is intended to convey the concept that changes in epigenetics can lead to changes in transcription and vice versa, which is a theme as evidence is discussed throughout the manuscript. We acknowledge that differences may exist in the way in which these changes respond to the degree and/or extent of the hypoxic stimulus. However, there is insufficient evidence to draw meaningful conclusions on this and whether a hierarchical response exists. We have discussed this in the manuscript (lines 406-412) in response to this and other comments. We have also added a statement in the discussion to acknowledge that the degree and extent to which the various processes are interlinked remains an open question (lines 680-681).
+ It would help the reader if at the end of the introduction a brief statement is made what this review specially will address.
A brief statement has been added (lines 67-68) as suggested.
+ Page 3, line 128 – the authors mention that on average 4 enhancers interact with a promoter through cohesion-mediated chromatin loops. This reviewer is not sure if this number is as established as the authors describe. In fact, the existence of enhancer-promoter loops as means to facilitate their interaction is not always observed. For instance, recently the Bickmore’s group (Benabdallah et al 2010 Mol Cell) recently showed that upon activation the enhancer and promoter were further separated, contradicting the chromatin loop formation for promoter-enhancer interaction.
The number of 4 was derived from the ENCODE Consortium analysis. We acknowledge that other analyses have arrived at slightly different estimates and have therefore amended the manuscript to be less precise as well as to reflect other possible mechanisms of enhancer-promoter interaction (line 131).
+ Page 4, line 130 – a recent paper by the Wysocka lab (Gu et al 2020 Mol Cell) very neatly describe how cohesin and transcription play opposing roles in CTCF cluster formation. It would be a nice addition if these recent results were briefly discussed in this paragraph.
Thank you for drawing our attention to this recent paper using super-resolution imaging to examine the effects of cohesion and transcription on CTCF cluster formation. We have added a reference to it in the manuscript (lines 132-133).
+ The first paragraph on Page 5 reads a little difficult. For instance, it is unclear what “this” on line 175 refers to. How does HIF bind to accessible HREs and DHS’s? Doesn’t a DHS mean it is accessible? Studies Schodel et al and Platt et al are given a lot of weight as if they present fully established dogma. The term “pioneer factor” is not introduced. Please, improve the clarity of this important paragraph. Maybe, the authors can suggest future experiments with CUT&Tag and/or single cell ChIP-seq to better understand cell-to-cell variability in hypoxic response.
We have amended the text in this paragraph to improve the clarity as suggested. We agree that new techniques such as CUT&RUN and CUT&Tag provide an exciting opportunity to study cell-to-cell variations in the hypoxic response and have added a discussion of this in the section on ChIP-seq analysis of histone modification (lines 515-518).
+ The second paragraph of Page 5 leaves the reader hanging how hypoxia can influence methylation.
Thank you. We felt that this question was better dealt with later in the manuscript and have indicated that we will return to it in the text (line 219).
+ The authors state that chromatin looping occurs (line 224), whereas in the preceding paragraph it is noted that very few higher-order chromatin structure studies have been done thus far. It might be more appropriate if the current state of affairs if summarized and the opinion or model of the authors is subsequently stated.
We have amended the manuscript to make it clear that the proposed model is our own interpretation of the evidence (line 247)
+ Page 6, line 240 – “the effects of oxygen and hypoxia” reads a bit weird. Oxygen is a molecule, whereas hypoxia is a biological state. Maybe the sentence can be rewritten to reflect the differences.
This has been changed to read “oxygen availability” (line 265)
+ Page 7, line 276-7 – It is noted that 2-oxogluterate-dependent dioxygenase has a linear enzymatic response to O2 concentrations, it is not noted what the biological relevance of this observation is. This is however noted for prolyl hydroxylase (lines 279-81). Maybe this can be added.
We have added a statement to clarify the relevance of this linear response (lines 308-309)
+ Page 8, line 311 – The linker histone H1.4 is introduced without explanation what a linker histone is or is thought to do.
We have added an explanation of the role of histone H1.4 (lines 369-370)
+ Table 2 – behind each v-mark, there appears to be an empty square. Is this supposed to be there? The font is also very small and difficult to read, especially in dim light conditions.
Unfortunately, we are unable to see the “empty squares”, referred to, in our version of the manuscript, but we will be sure to check the final proofs carefully to make sure that they do not appear in the published version. We have now increased the size of the font used in this table.
+ In section 6 a slew of chromatin studies are mentioned, but to this reviewer, what I am missing is a sense of how quick these changes were observed when cells were subjected to hypoxia conditions (even though they are reported in Table 4). In the hormone-gene expression field, it is widely expected that there are fast transcriptional responses directly induced by hormone receptors binding to genomic sites and subsequent medium and slow responsive genes. It would be informative if such data, as far as known, was clarified, as this would provide a mechanistic understanding in which order what event might occur.
We agree with the reviewer that it is difficult to get a sense of how quickly global changes in individual histone modifications are occurring in hypoxia. We have attempted to redress this by including the duration of hypoxia used in each study in table 3 as well as in table 4. However, as can clearly be seen, the duration of hypoxia used varies extensively from 30 minutes to 14 weeks. Furthermore, there is clearly a publication bias in these results as very few groups have reported unchanged histone modifications at the timepoints they used. We have added a statement to the manuscript alerting the reader to this important issue (lines 406-412). We also agree with the reviewer that this is an important open question and possibly one that would be best addressed at a locus-specific level using newer pangenomic approaches. We have therefore suggested this as an important area for future study in the section on locus-specific changes in histone modification (lines 496-500).
+ Page 12, line 416 – “2” should be “Two”.
This has been changed as suggested (line 506)
+ Page 12, last paragraph – it is confusing to this reviewer how at first the authors caution for overinterpretation of linking hypoxia-associated changes in histone modifications and gene expression, to end the paragraph with a HIF and RNAP2 to directly alter the histone modification landscape. Maybe it worth considering rewriting this paragraph to more clearly reflect the thought and concerns.
The concept that we wished to convey in this paragraph was the possibility that causality could operate in either direction. We had included the word “might” in the final sentence to emphasise this as a possibility rather than as established fact. However, we have modified the text to make this clearer (line 527).
+ Section 9 ends rather abruptly. Maybe a concluding sentence/paragraph could be helpful for the reader.
We have added a concluding sentence to this paragraph to summarize this section as suggested (lines 599-601).
+ Figure 3 – to this reviewer it is puzzling to see histone marks being added and removed from a cartoon of DNA that depleted of nucleosomes. Also, the variable effects of acetylation on HIF1a stability does not shine through. The figure suggests that K532, K674, and K709 first have to be acetylated and right away deacetylated, as well as methylated for it to be ultimately degraded. It is not clear that EGLN3 is a negative regulator of HIF1a.
We have amended Figure 3 in the light of these comments
+ In section 11, the authors discuss the potential of epigenetic “memory”. This paragraph is confusing how different findings are brought up and under which premise they are presented. Are HIF-responsive genes supposed to respond like cell-type specific marks? Are cultured cell lines directly comparable to mice studies? It might be worth rewriting this paragraph to clearly reflect what was found in which system and under which conditions, and how these findings differ or agree.
This section is intended to discuss the possibility that epigenetic changes induced by hypoxia may persist following re-oxygenation and affect the transcriptional response to subsequent repeated hypoxic insult. We have amended the wording to make this concept clearer (lines 644-660).
Reviewer 3 Report
This is very solid review addressing the role of chromatine modification in the process of HRE motives recognition by HIFs.
The subject is not well represented in literature and thus this work is timely and valid.
I have just couple of comments:
- Authors focus mainly on cancer cells - these has numerous abbreviations (adaptations) both to hypoxia and against apoptosis. Hence the concluding based on cancer cell lines is limited. I think authors should mention this problem.
- The HIF1 and HIF2 are not equivalent, and recent data are published from primary human epithelial cells regarding HRE and genes that are HIF1 and HIF2 dependent. Please see https://pubmed.ncbi.nlm.nih.gov/30917010/
- The phd (egln3) is induced by HIF1 but on mRNA levels !!!! the protein levels remain constant. Hence its hard to speak about negative feedback. Its being made in order to quickly inactivate HIF up on oxygen restoring, but first HIF1AN is preventing HIF transcriptional activity
- The HIF levels decline during prolonged hypoxia because of 2 main reasons : decline in HIF mRNA levels https://pubmed.ncbi.nlm.nih.gov/30917010/
and global inhibition of translation during prolonged hypoxia :https://www.tandfonline.com/doi/pdf/10.4161/cbt.3.6.1010
and https://www.sciencedirect.com/science/article/abs/pii/S1084952105000546
Author Response
This is very solid review addressing the role of chromatine modification in the process of HRE motives recognition by HIFs.
The subject is not well represented in literature and thus this work is timely and valid.
I have just couple of comments:
- Authors focus mainly on cancer cells - these has numerous abbreviations (adaptations) both to hypoxia and against apoptosis. Hence the concluding based on cancer cell lines is limited. I think authors should mention this problem.
This is an important point and we have added a comment in the final discussion (pages 665-666) to draw the reader’s attention to it.
- The HIF1 and HIF2 are not equivalent, and recent data are published from primary human epithelial cells regarding HRE and genes that are HIF1 and HIF2 dependent. Please see https://pubmed.ncbi.nlm.nih.gov/30917010/
The reviewer correctly highlights the different roles of HIF-1 and HIF-2 for which there is a lot of evidence in the literature. This is an important issue and one that could form the focus of a review in itself. However, we felt that a full discussion of this matter was beyond the scope of this review to cover this issue in detail. Nevertheless, we have amended the manuscript to emphasise this important point and have directed readers to a review by the Simon group that covers this area as well as the recent paper mentioned above (lines 150-153).
- The phd (egln3) is induced by HIF1 but on mRNA levels !!!! the protein levels remain constant. Hence its hard to speak about negative feedback. Its being made in order to quickly inactivate HIF up on oxygen restoring, but first HIF1AN is preventing HIF transcriptional activity
Thank you. We agree that PHD3/EGLN3 is clearly a transcriptional target of HIF (mRNA levels are induced in hypoxia and altered by genetic intervention on HIF-a and there is a HIF-binding site at the EGLN3 locus). However, it has clearly been shown that this also results in hypoxic induction of PHD3 protein (e.g. Appelhoff et al JBC 2004 as cited in the paper by Bartoszewski below). Furthermore, overexpression of PHD3 suppresses HIF-1a protein expression in hypoxia (e.g. Cioffi et al BBRC 2003 as cited in the paper by Liu et al below). Taken together, this indicates that transcriptional activation of EGLN3 by HIF can negatively impact the HIF pathway in hypoxia in addition to any role that this may play during re-oxygenation.
- The HIF levels decline during prolonged hypoxia because of 2 main reasons: decline in HIF mRNA levels https://pubmed.ncbi.nlm.nih.gov/30917010/ and global inhibition of translation during prolonged hypoxia: https://www.tandfonline.com/doi/pdf/10.4161/cbt.3.6.1010 and https://www.sciencedirect.com/science/article/abs/pii/S1084952105000546
We agree that, in addition to the above, multiple mechanisms oppose the activation of the HIF pathway in hypoxia. Indeed, the reduction of HIF-1a mRNA (thought to be due, at least in part, to a HIF-dependent anti-sense transcript at the HIF-1a gene locus) and the global reduction in translation during hypoxia are well described in the literature as indicated by the reviewer. It is likely that these have different time courses of action and differential effects on the different HIF-a isoforms that result in the differential duration of HIF-1a and HIF-2a protein induction described by Bartoszewski and others. However, it was felt that a detailed discussion of this was outside of the scope of this manuscript, which focusses on transcription and chromatin in hypoxia.